# SlotFM: A Motion Foundation Model with Slot Attention for Diverse Downstream Tasks

## Abstract

Wearable accelerometers are used for a wide range of applications, such as gesture recognition, gait analysis, and sports monitoring. Yet most existing foundation models focus primarily on classifying common daily activities such as locomotion and exercise, limiting their applicability to the broader range of tasks that rely on other signal characteristics. We present SlotFM[1], an accelerometer foundation model that generalizes across diverse downstream tasks. SlotFM uses Time-Frequency Slot Attention, an extension of Slot Attention that processes both time and frequency representations of the raw signals. It generates multiple small embeddings (slots), each capturing different signal components, enabling task-specific heads to focus on the most relevant parts of the data. We also introduce two loss regularizers that capture local structure and frequency patterns, which improve reconstruction of fine-grained details and helps the embeddings preserve task-relevant information. We evaluate SlotFM on 16 classification and regression downstream tasks that extend beyond standard human activity recognition. It outperforms existing self-supervised approaches on 13 of these tasks and achieves comparable results to the best performing approaches on the remaining tasks. On average, our method yields a 4.5% performance gain, demonstrating strong generalization for sensing foundation models.

## 1 Introduction

Advances in self-supervised learning (SSL) and large-scale datasets have enabled foundation models that support multiple tasks through shared representations (Yang et al., 2024; Oquab et al., 2023). This is particularly valuable for wearable devices, where maintaining separate models dedicated for each task is often impractical due to memory and compute constraints. Accelerometers are widely used sensors in wearables for diverse motion-related tasks. Recent studies show that SSL approaches can train foundation models effective in Human Activity Recognition (HAR) tasks such as exercise and locomotion classification (Logacjov, 2024). However, their applicability to broader accelerometer tasks, such as gait analysis and gesture recognition, remains largely unexplored. This contrasts with domains such as audio, where foundation models have been applied beyond a single task, spanning speech-to-text, speaker identification, and emotion recognition.

While certain SSL methods may work well for specific domains of accelerometer tasks, they may not generalize to a broader range of tasks. Augmentation-based methods like SimCLR (Tang et al., 2020) train models to be invariant or sensitive to specific signal features. For example, rotation augmentations improve robustness to device placement, benefiting activity recognition (Xu et al., 2025). Yet the same invariance can hinder tasks like gait analysis, where orientation carries meaningful information. Thus, training a foundation model to be biased toward a particular signal characteristic may restrict its generalization. This motivates a reconstruction-based approach that preserves the full input signal. However, standard autoencoders compress each window into a single embedding, which limits their ability to capture the diverse temporal and spectral patterns needed across tasks.

To address the challenge of building a foundation model that generalizes across diverse accelerometer tasks, we propose Time-Frequency Slot Attention, a novel SSL approach that extends Slot Attention (Locatello et al., 2020) to operate over both the time and frequency domains. Time-Frequency

---

[1]Code implementation can be found at *(will be updated after acceptance)*

Figure 1: Overview of SlotFM. Accelerometer signals are decomposed into multiple frequency bands. Time-Frequency Slot Attention produces multiple slot vectors that capture distinct components of the signal. Task-specific heads attend to these slot for diverse downstream tasks.

Slot Attention encodes each signal into multiple vectors, or "slots", which are then decoded to jointly reconstruct the original input. This encourages the slots to collectively preserve the full signal content while individually capturing distinct learned components. Using these slots as the embedding for downstream tasks then enables task-specific heads to attend more effectively to relevant features, improving generalization across tasks. Additionally, we introduce two loss regularizers: the Structural Similarity Index Measure (SSIM), inspired by the image domain, to encourage the embeddings to capture structural patterns of the signal, and the Multi-Scale Short-Term Fourier Transform (MS-STFT), inspired by the audio domain, to emphasize high-frequency details.

We evaluate our model, SlotFM, on 11 diverse downstream tasks, beyond those typically considered in foundation model studies, spanning both classification and regression across daily activities, sports, gestures, and transportation. We also test on five tasks of an existing Human Activity Recognition (HAR) benchmark. SlotFM outperforms state-of-the-art SSL methods on 13 out of 16 downstream tasks and achieves comparable results to the best performing approaches on the remaining tasks. Overall, SlotFM achieves an average improvement of 4.5% and in some cases even surpasses fully supervised models. Moreover, while baseline methods fluctuate in performance across tasks, SlotFM performs consistently well. Finally, analysis of head weights shows that certain slots are emphasized more than others depending on the task, demonstrating the adaptability of slot-based embeddings. These results demonstrate the strength of our approach in creating a single foundation model that produces embeddings generalizable to distinctively different target tasks.

Our key contributions are as follows:
**1.** We propose Time-Frequency Slot Attention, a new self-supervised learning approach that decomposes raw accelerometer signals into distinct slot-based embeddings, along with two loss regularizers that improve signal reconstruction of finer details.
**2.** We train SlotFM, an accelerometer foundation model using our approach, and evaluate it on 16 diverse downstream tasks, collected exclusively from publicly available datasets.
**3.** We release the code for our model training and downstream benchmark setup to support reproducibility and future research.

## 2 RELATED WORK

**Foundation models for accelerometer data:** Several works have proposed foundation models using accelerometer data over the past few years, using various SSL approaches to train large backbone models on unlabeled accelerometer datasets. One common approach is to design pretext tasks using augmentations (Saeed et al., 2019; Yuan et al., 2024), and train the model to predict which augmentations were applied. Contrastive learning has also been widely used, typically by applying augmentations that are assumed not to change the meaning of the signal, and treating them as positive pairs while using all other signals as negatives (Tang et al., 2020). REBAR improves on this by

learning a distance model that measures the similarity between sequences based on reconstruction error when one is reconstructed using information from the other (Xu et al., 2024). RelCon builds on this idea with a contrastive learning method that models relative differences using soft positive and negative pairs (Xu et al., 2025). Another class of methods trains the model to reconstruct the input signal, such as autoencoders (Haresamudram et al., 2019) or masked reconstruction (Logacjov & Bach, 2024). These methods are useful because they require the model to preserve all signal information in the embedding to reconstruct the full input.

**Foundation models for other wearable sensor data:** Beyond accelerometers, many foundation models have been proposed using other wearable sensors, including gyroscopes, magnetometers, altimeters (Narayanswamy et al., 2025; Xu et al., 2021), physiological signals like PPG and ECG (Abbaspourazad et al., 2024a;b; Luo et al., 2024; Pillai et al., 2025), and higher-level behavioral signals (Erturk et al., 2025). These typically follow similar SSL strategies such as masked reconstruction. Multi-modal approaches can leverage natural pairings between modalities to define positive pairs for contrastive learning (Liu et al., 2023; Deldari et al., 2024) or to distill knowledge from one modality to another (Abbaspourazad et al., 2024b). Mixtures of such training schemes are also common, as in the work of Das et al. (2025), which combines self-supervision, multimodal supervision with parallel text and video data, and nearest-neighbor supervision. Some works also propose signal-specific methods, such as PaPaGei, which incorporates PPG-derived metrics like the stress-induced Vascular Response Index into training (Pillai et al., 2025).

**Accelerometer FM task diversity:** Most accelerometer foundation models have focused on human activity recognition tasks, such as exercises or daily routines (Koskimäki et al., 2017; Blunck & Dey, 2015). Some datasets cover more specialized tasks like shorter actions or freeze of gait (Scholl et al., 2015; Bachlin et al., 2009), and RelCon extended evaluation to regression tasks for gait metrics such as double support time and stride velocity (Xu et al., 2025). Yet in practice, accelerometers and IMUs have been used for a far broader range of tasks, including pose estimation (Mollyn et al., 2023), sports action classification (Park et al., 2024), hand gesture recognition (Zhang et al., 2021), jump height estimation (Villa et al., 2024), and user identification through gait (Liberman* et al., 2024). This indicates a wider space where accelerometer foundation models could be applied and evaluated. Recent work on foundation models for wearable data has been shifting toward more general-purpose models. Erturk et al. (2025) utilized higher-level behavioral data from wearables to build a foundation model effective on 57 diverse health-related tasks. In the PPG domain, Pillai et al. (2025) proposed a foundation model tested on 20 diverse classification and regression tasks.

## 3 METHODOLOGY

### 3.1 TIME-FREQUENCY SLOT ATTENTION

Slot Attention is an attention mechanism that encodes input data into a fixed number of vectors called "slots" (Locatello et al., 2020). When trained with a self-supervised reconstruction objective, similar to an autoencoder, these slots compete to capture different parts of the input. For example, when Slot Attention is applied to images, each slot captures a distinct object in the image. The core mechanism is the cross-attention between input embeddings and slot vectors. At each forward

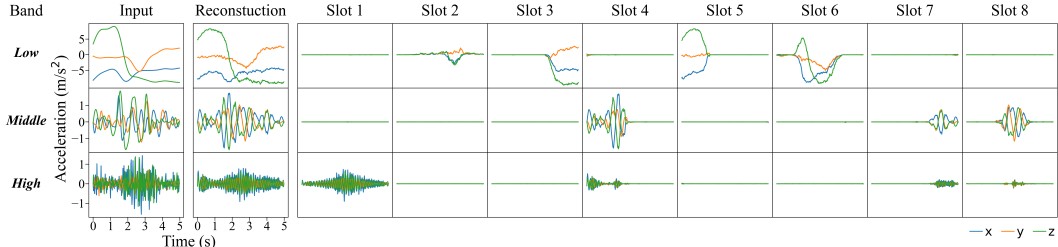

Figure 2: Slot reconstructions when bandpassed signals are used as input. The eight slots distribute across both time and frequency dimensions, capturing distinct components of the input signal.

pass, $K$ initial slot vectors are first sampled from a learned normal distribution. Then, the cross-attention matrix between these slots and the input embeddings are used to compute updates for the slot vectors, which is repeated for $U$ iterations to produce the final slot vectors. Thus, the initial slot vectors determine which parts of the input to focus on, acting as "queries" that attend over it.

We adapt Slot Attention to raw accelerometer time series so that each slot captures a different segment of a signal window. In the original algorithm, however, the initial slots are resampled at every forward pass, and since they determine which part of the input each slot attends to, the assignment of information across slots changes from one pass to the next. For example, a region of the signal captured by the first slot in one pass may instead be captured by the second slot in the next, even for the same input. This variability makes it difficult to use the slots directly as embeddings for downstream tasks, as the ordering of information is inconsistent. To address this, we replace resampling with fixed learnable vectors: we initialize $S_{\text{initial}} \in \mathbb{R}^{K \times D_{\text{slot}}}$ once as trainable parameters, and at each forward pass we use $S_{\text{initial}}$ as the initial value of the slots $S$. This ensures that the slots maintain a consistent semantic order and can be more reliably used by task-specific heads.

Since motion signals contain information across varying frequencies, spectrograms or frequency-filtered signals have often been used to decompose the signal along this dimension. (Yang & Hsu, 2010). To encourage slots to capture frequency-specific aspects of the input, we use a 4th order Butterworth bandpass filter to decompose the original signal into three non-overlapping bands: 0-1 Hz, 1-4 Hz, and above 4 Hz, following prior work on human movement frequency (Fridolfsson et al., 2019). While finer decompositions or full spectrograms could provide more granular information, our experiments show that three bands offer a good balance between computational cost and signal detail. Figure 2 visualizes the signal reconstructions of 8 slots when bandpassed signals are used as input. The slots attend to different regions in both the temporal and frequency dimensions, whereas when the original signal is used as a single band, they capture only different temporal segments. Further analysis of the impact of the number of bands is provided in the Appendix A.4.

## 3.2 MODEL ARCHITECTURE

Figure 3 illustrates the SlotFM architecture. The input signal $X_{\text{in}}$ is first decomposed into three frequency bands ($X_{\text{low}}$, $X_{\text{mid}}$, and $X_{\text{high}}$). Each band is processed by a separate ResNet-style encoder that preserves the full temporal length. Encoder weights are not shared, since each band captures distinct characteristics. To provide positional context, we add soft positional embeddings using learned 2D coordinate features. This results in the concatenated sequence of encoded frames $E = [E_{\text{low}}; E_{\text{mid}}; E_{\text{high}}]$. Slot Attention is applied between the encoded frames $E$ and the slot vectors $S$ to produce an attention matrix, which is normalized and used to compute slot updates as the weighted

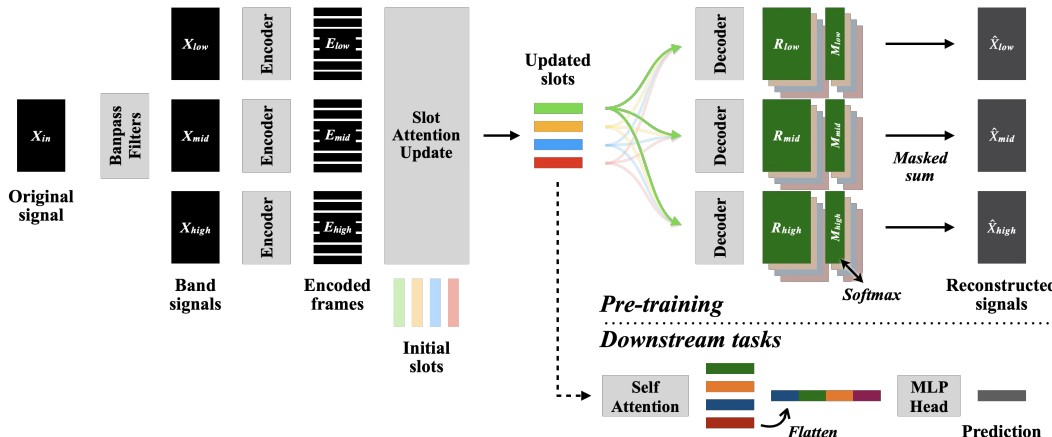

Figure 3: Architecture of SlotFM. The input accelerometer signal is split into three frequency bands, encoded with convolutional encoders, and aggregated using Slot Attention into slot vectors. For pre-training, the slots are decoded back into each band to reconstruct the original signal. For downstream tasks, the slots are processed with a self-attention layer and an MLP head for prediction.

mean of the encoded frames. Each update is passed through a GRU cell and a multi-layer perceptron, as in the original method, to generate new slot vectors. This cross-attention and update process is repeated $U$ times to form the final slot vectors.

During pre-training, each slot is decoded back into the three bands using band-specific ResNet decoders. Each decoder outputs a reconstructed signal $R$ and a temporal mask $M$ for each slot. For each band, the masks from all slots are normalized across slots using a softmax, and the resulting weights are used to blend the per-slot reconstructions into the final output $\hat{X}$ for that band. The full pre-training procedure is provided in the pseudocode in Appendix A.2.1.

For downstream tasks, the decoders are removed and the final slot vectors are used directly as embeddings. A single multi-head self-attention layer captures inter-slot relationships, after which the slots are flattened, concatenated, and passed to an MLP head for prediction.

## 3.3 RECONSTRUCTION LOSS

Most existing reconstruction-based self-supervised learning methods for IMU signals use Mean Squared Error (MSE) to compute the loss between the original and reconstructed signals. However, this can be problematic for signals like accelerometers that contain high-frequency motion, as even a slight shift in prediction of a peak can lead to a large loss (Cuturi & Blondel, 2017; Mathieu et al., 2015). As a result, models tend to make conservative, smoothed predictions that capture only low-frequency trends, losing the fine-grained motion details crucial for some downstream tasks.

To address this, we introduce two regularization losses to use alongside MSE. The first is the Structural Similarity Index Measure (SSIM), originally used in the image domain to compare local structure by measuring luminance and contrast instead of individual pixel values (Zhao et al., 2016). We adapt it to raw signals by computing the mean and variance over short segments with a sliding window to capture structural patterns across the input. The SSIM loss function is defined as follows:

$$\mathcal{L}_{\text{SSIM}} = 1 - \text{SSIM}(x, y),$$

$$\text{SSIM}(x, y) = \frac{(2\mu_x\mu_y + C_1)(2\sigma_{xy} + C_2)}{(\mu_x^2 + \mu_y^2 + C_1)(\sigma_x^2 + \sigma_y^2 + C_2)},$$

where $x$ and $y$ are the original and reconstructed signals, $\mu_x, \mu_y$ are their local means, $\sigma_x^2, \sigma_y^2$ are their local variances, $\sigma_{xy}$ is the local covariance, and $C_1, C_2$ are small constants to stabilize division.

To further encourage the embedding to retain high-frequency variations, we incorporate the multi-scale short-term Fourier transform (MS-STFT) loss from audio signal processing work (Défossez et al., 2023). Unlike a single-resolution STFT loss, the MS-STFT computes magnitude spectra at multiple window sizes, allowing the model to capture both fine-grained high-frequency details and longer-term low-frequency structures. At each scale, we compute a combination of L1 and MSE losses between the magnitude spectra of the original and reconstructed signals, and average them across all scales. The MS-STFT loss is defined as follows:

$$\mathcal{L}_{\text{MS-STFT}}(x, y) = \frac{1}{|\mathcal{F}|} \sum_{n_{\text{fft}} \in \mathcal{F}} \Big( \text{MAE}(|\text{STFT}_{n_{\text{fft}}}(x)|, \ |\text{STFT}_{n_{\text{fft}}}(y)|)$$

$$+ \text{MSE}(|\text{STFT}_{n_{\text{fft}}}(x)|, \ |\text{STFT}_{n_{\text{fft}}}(y)|) \Big),$$

$$\mathcal{F} = \{16, 32, 64, 128\}.$$

where $x$ and $y$ are the original and reconstructed signals, $|\cdot|$ denotes the magnitude spectrum, $\text{STFT}_{n_{\text{fft}}}$ is the short-time Fourier transform, $\mathcal{F}$ is the set of FFT sizes, and MAE/MSE are the mean absolute and mean squared errors. We use a Hann window of length $n_{\text{fft}}$ and hop length $n_{\text{fft}}/4$.

Because each band captures different patterns, we use different loss combinations for each one. The low-frequency band uses only $\mathcal{L}_{\text{MSE}}$, the mid-frequency band uses all three losses ($\mathcal{L}_{\text{MSE}}$, $\mathcal{L}_{\text{SSIM}}$, $\mathcal{L}_{\text{MS-STFT}}$), and the high-frequency band uses only $\mathcal{L}_{\text{SSIM}}$ and $\mathcal{L}_{\text{MS-STFT}}$. We define the total loss as a weighted combination of all three terms:

$$\mathcal{L}_{\text{total}} = \alpha \cdot \mathcal{L}_{\text{MSE}}(x, y) + \beta \cdot \mathcal{L}_{\text{SSIM}}(x, y) + \gamma \cdot \mathcal{L}_{\text{MS-STFT}}(x, y),$$

where the weights $\alpha$, $\beta$ and $\gamma$ are hyperparameters.

## 4 EXPERIMENTS

### 4.1 PRE-TRAINING

We trained models on CAPTURE-24 (Chan et al., 2024), the largest open-source IMU dataset, containing real-world wrist-worn activity tracker data. The pre-training set includes approximately 2,500 hours of data from 151 participants, with 120 used for training and the remaining 31 for validation. Following Yuan et al. (2024), we apply weighted sampling during training, where windows are sampled in proportion to their standard deviation. This gives higher priority to high-movement windows, which is especially important since in-the-wild motion data often includes long periods of low activity that are less informative for model training.

The model takes 5-second windows of raw 3-axis accelerometer data sampled at 50Hz as input. We train the model for 500 epochs using four V100 GPUs, with early stopping on validation triggered after 40 epochs. The encoder and decoder networks follow a ResNet-style architecture, each consisting of four residual blocks built with 1D convolutions (kernel size 3) for each band. We use eight slot vectors of dimension 32, resulting in a combined embedding of size 256. The total model size of the encoder is 4.8 million parameters. Further implementation details are provided in Appendix A.2.

### 4.2 DOWNSTREAM EVALUATION

To evaluate how well our foundation model's learned embedding generalizes across diverse downstream tasks, we conduct two sets of evaluations. In the Task Diversity Evaluation, we compare our SSL approach to existing methods on 11 curated downstream tasks. These tasks span various domains, including gesture and exercise classification, as well as ball throw speed estimation. In the HAR Benchmark Evaluation, we compare our model to an existing benchmark on human activity recognition, showing how well it performs against standardized models and tasks.

#### 4.2.1 TASK DIVERSITY EVALUATION

We compiled a set of 11 downstream tasks from open-source datasets. These tasks span diverse applications where accelerometers have been used in prior work but remain largely unexplored in the context of foundation models. Table 1 provides an overview of each downstream task, with more details on preprocessing found in Appendix A.1.

Table 1: Downstream tasks and datasets used for this benchmark. For classification tasks, the number of classes is indicated. Details on dataset parsing and task design are provided in Appendix A.1.

| | Task name | Task description | # Subjects | # Samples | Dataset |
|---|---|---|---|---|---|
| **Classification** | Basketball | Basketball actions (e.g., layup, shoot). [C=5] | 24 | 9781 | Hoelzemann et al. (2023) |
| | Cooking | Cooking actions (e.g., turn stove on, wipe pan). [C=14] | 17 | 237 | Arakawa et al. (2023) |
| | Locomotion | Exercise and locomotion (e.g., walking, jogging). [C=8] | 15 | 11527 | Sztyler & Stuckenschmidt (2016) |
| | TennisShot | Tennis shot measured from the non-dominant arm. [C=6] | 20 | 6000 | Park et al. (2024) |
| | Transportation | Transportation mode that participant is in. [C=6] | 3 | 40059 | Gjoreski et al. (2018) |
| | Workouts | Outdoor workouts (e.g., burpees, lunges). [C=18] | 24 | 8008 | Bock et al. (2024) |
| | Writing | Letters/characters written on a device. [C=39] | 10 | 11685 | Roggen (2022) |
| **Regression** | JumpPower | Power of vertical jump | 73 | 1128 | White et al. (2022) |
| | NumSteps | Number of steps taken during the given window | 25 | 492 | Santos et al. (2022) |
| | ThrowSpeed | Speed of handball throw | 4 | 105 | Gençoğlu & Gümüş (2020) |
| | WalkDistance | Distance walked during the given window | 25 | 492 | Santos et al. (2022) |

We compare our training approach against five competitive self-supervised learning methods. All models are trained on the same dataset with identical configurations. The encoder is a ResNet-style network with 12 residual blocks, matching the model size and embedding dimension of SlotFM. Baselines include Autoencoder (Haresamudram et al., 2019), Masked Autoencoder (Haresamudram et al., 2020), and SimCLR (Tang et al., 2020), commonly used in accelerometer foundation models, as well as Augmentation Prediction (Yuan et al., 2024) and RelCon (Xu et al., 2025), recent approaches that outperform prior baselines. We also include a supervised model where the full encoder architecture is trained from scratch for each individual downstream task. Implementation and parameter details for all baselines are provided in Appendix A.3.

All self-supervised models are first pre-trained on the CAPTURE-24 dataset using their respective methods. The backbone is then frozen, and a lightweight MLP head is trained for each downstream task. For the supervised model, the full model is trained end-to-end for each task, using two additional lower learning rates to reflect the model size differences. The hidden dimensions of the baseline heads are set so that the total number of parameters is similar to that of SlotFM.

All tasks are evaluated using five folds, with 20% of participants held out for testing in each fold and the remaining 80% split 80/20 into training and validation. Participant assignments per fold are kept consistent across all baselines to ensure fair comparison. We use Cross Entropy Loss for classification tasks, applying class weights in cases of high imbalance, and Mean Squared Error for regression tasks. All downstream training is run for 200 epochs with early stopping after 10 epochs.

### 4.2.2 HAR Benchmark Evaluation

Haresamudram et al. (2022) provides a benchmark for evaluating self-supervised learning methods on human activity recognition using accelerometer data. They train various SSL methods on the CAPTURE-24 dataset and evaluate them on multiple HAR datasets collected from different body locations. Among the benchmark's evaluation protocols, we follow their MLP classifier setup, where the SSL backbones are frozen and three linear layers with batch normalization, dropout, and ReLU are trained for each downstream task. We match their configuration, including the 2-second window length, 50 Hz sampling rate, and 128-dimensional embedding size. We follow the same 5-fold split protocol, though the original folds are not available and the exact participant assignments may differ. To account for the smaller window size, we use four slot vectors, each of dimension 32. We compare our results directly with the published benchmark results on 5 different commonly used HAR datasets: HHAR (Stisen et al., 2015), USC-HAD (Zhang & Sawchuk, 2012), Motion-sense (Malekzadeh et al., 2019), PAMAP2 (Reiss, 2012), and FoG (Bachlin et al., 2009), covering various target sensor locations that differ from the wrist location used in pre-training.

## 5 Results

### 5.1 Task Diversity Evaluation Results

Table 2 shows the performance of each SSL method on the 11 diverse downstream tasks. Classification performance is measured using macro F1 score and regression performance is measured using RMSE. SlotFM outperforms all SSL baselines across all tasks except the Transportation and WalkDistance tasks, where it trails Augmentation Prediction and RelCon, respectively, by only a small margin. In five tasks, SlotFM even surpasses the Supervised model, highlighting the benefit of pre-training on large datasets, as the model can learn fundamental knowledge of the data domain that helps extract meaningful embeddings. Note that performance standard deviations are relatively high, mainly due to imbalanced data sizes and missing classes across participants.

For some tasks, certain SSL baselines achieve results close to SlotFM and outperform the other baselines. The strongest baseline, however, is different depending on the task. For example, Augmentation Prediction performs similarly to SlotFM on the Workouts task, Autoencoder on the Writing task, and SimCLR and RelCon on the ThrowSpeed and WalkDistance tasks. But on other tasks, these same baselines show a large drop. This suggests that the embeddings that these baselines produce do not generalize well across tasks with diverse characteristics. In contrast, SlotFM maintains consistently strong performance across all tasks, showing the generalizability of its embeddings.

Table 2: Downstream task results for SlotFM compared to supervised baselines and existing self-supervised methods. Classification performance is reported with F1 score (higher is better) and regression performance with RMSE (lower is better). Best performing models are indicated in bold.

| Tasks | Supervised | Autoencoder | Masked Autoencoder | Augmentation Prediction | SimCLR | RelCon | SlotFM |
|---|---|---|---|---|---|---|---|
| **Classification - F1 ($\uparrow$)** | | | | | | | |
| Basketball | $59.9 \pm 15.3$ | $57.1 \pm 13.2$ | $54.1 \pm 12.6$ | $47.8 \pm 8.1$ | $53.3 \pm 8.7$ | $54.3 \pm 9.2$ | $\mathbf{64.7 \pm 12.2}$ |
| Cooking | $\mathbf{51.2 \pm 11.7}$ | $50.1 \pm 14.7$ | $45.5 \pm 11.9$ | $28.5 \pm 10.0$ | $48.2 \pm 10.1$ | $48.1 \pm 11.0$ | $50.9 \pm 12.2$ |
| Locomotion | $72.5 \pm 8.3$ | $66.4 \pm 10.2$ | $74.2 \pm 9.0$ | $72.1 \pm 6.9$ | $70.0 \pm 7.6$ | $70.7 \pm 8.3$ | $\mathbf{74.6 \pm 9.2}$ |
| TennisShot | $78.2 \pm 15.0$ | $75.8 \pm 12.4$ | $68.4 \pm 13.7$ | $60.6 \pm 12.2$ | $64.3 \pm 14.1$ | $63.4 \pm 16.6$ | $\mathbf{81.5 \pm 11.5}$ |
| Transportation | $56.2 \pm 7.4$ | $55.8 \pm 3.4$ | $57.6 \pm 4.7$ | $\mathbf{63.6 \pm 7.7}$ | $56.9 \pm 6.4$ | $56.4 \pm 3.0$ | $63.4 \pm 4.8$ |
| Workouts | $\mathbf{71.7 \pm 12.4}$ | $66.5 \pm 13.0$ | $68.4 \pm 13.6$ | $69.3 \pm 9.7$ | $63.8 \pm 9.6$ | $64.8 \pm 11.2$ | $69.9 \pm 11.3$ |
| Writing | $\mathbf{52.2 \pm 14.9}$ | $46.3 \pm 13.1$ | $34.6 \pm 11.5$ | $12.4 \pm 1.7$ | $24.9 \pm 7.9$ | $31.0 \pm 9.9$ | $48.1 \pm 11.6$ |
| **Regression - RMSE ($\downarrow$)** | | | | | | | |
| JumpPower | $\mathbf{3.61 \pm 1.66}$ | $9.61 \pm 4.78$ | $7.32 \pm 3.24$ | $9.58 \pm 2.78$ | $9.63 \pm 2.99$ | $7.84 \pm 3.66$ | $\mathbf{6.32 \pm 2.69}$ |
| NumSteps | $0.61 \pm 0.29$ | $0.67 \pm 0.28$ | $0.59 \pm 0.18$ | $0.68 \pm 0.28$ | $0.59 \pm 0.28$ | $0.65 \pm 0.29$ | $\mathbf{0.57 \pm 0.26}$ |
| ThrowSpeed | $\mathbf{2.78 \pm 0.69}$ | $3.72 \pm 0.46$ | $3.67 \pm 0.60$ | $4.31 \pm 0.54$ | $2.95 \pm 0.32$ | $3.13 \pm 0.50$ | $\mathbf{2.92 \pm 0.41}$ |
| WalkDistance | $\mathbf{0.23 \pm 0.12}$ | $0.30 \pm 0.15$ | $0.27 \pm 0.11$ | $0.28 \pm 0.13$ | $0.26 \pm 0.14$ | $\mathbf{0.25 \pm 0.13}$ | $0.27 \pm 0.11$ |

## 5.2 HAR BENCHMARK EVALUATION RESULTS

Table 3 compares SlotFM with SSL methods benchmarked in Haresamudram et al. (2022). SlotFM shows strong generalization, performing well even when the target task's sensor location differs from the pre-training location (wrist). It outperforms self-supervised and fully-supervised models on four of the five tasks. On Motionsense, SimCLR performs better by only a small margin.

Table 3: Performance of SlotFM on the Accel Benchmarking Study. Best models are shown in bold.

| | | HHAR (Wrist) | USC-HAD (Waist) | Motionsense (Waist) | PAMAP2 (Leg) | FoG (Leg) |
|---|---|---|---|---|---|---|
| | **SlotFM (Ours)** | $\mathbf{67.4 \pm 2.6}$ | $\mathbf{60.1 \pm 2.3}$ | $85.5 \pm 2.8$ | $\mathbf{65.6 \pm 3.3}$ | $\mathbf{57.7 \pm 2.2}$ |
| Frozen SSL Backbone + MLP classifier | Multi-task self. sup | $57.5 \pm 1.9$ | $56.6 \pm 1.3$ | $83.2 \pm 1.3$ | $58.5 \pm 3.0$ | $54.2 \pm 1.1$ |
| | Masked Recons. | $55.0 \pm 2.6$ | $45.1 \pm 0.9$ | $75.7 \pm 1.9$ | $55.1 \pm 1.0$ | $52.5 \pm 1.0$ |
| | CPC | $58.1 \pm 1.1$ | $51.4 \pm 2.4$ | $84.7 \pm 1.1$ | $52.2 \pm 2.0$ | $51.2 \pm 1.0$ |
| | Autoencoder | $54.3 \pm 2.0$ | $51.3 \pm 2.2$ | $80.7 \pm 1.7$ | $56.9 \pm 2.0$ | $53.1 \pm 0.9$ |
| | SimCLR | $58.6 \pm 2.3$ | $53.7 \pm 4.1$ | $\mathbf{85.6 \pm 2.5}$ | $60.2 \pm 2.3$ | $52.5 \pm 1.4$ |
| | SimSiam | $54.7 \pm 1.3$ | $53.6 \pm 2.2$ | $83.4 \pm 1.6$ | $59.6 \pm 4.1$ | $50.4 \pm 1.7$ |
| | BYOL | $51.7 \pm 2.3$ | $51.0 \pm 2.4$ | $82.2 \pm 1.2$ | $55.8 \pm 1.3$ | $51.1 \pm 1.7$ |
| Fully supervised | DeepConvLSTM | $54.4 \pm 2.3$ | $53.6 \pm 0.5$ | $84.6 \pm 0.9$ | $51.2 \pm 1.9$ | $53.7 \pm 2.6$ |
| | GRU classifier | $46.0 \pm 2.1$ | $55.2 \pm 1.1$ | $87.1 \pm 0.9$ | $54.2 \pm 1.2$ | $54.0 \pm 1.1$ |
| | Conv. classifier | $55.4 \pm 1.2$ | $57.9 \pm 0.6$ | $\mathbf{89.3 \pm 0.5}$ | $59.8 \pm 1.5$ | $53.4 \pm 0.9$ |
| | MLP. classifier | $53.1 \pm 0.8$ | $55.6 \pm 1.1$ | $84.5 \pm 0.4$ | $50.0 \pm 0.4$ | $49.5 \pm 1.2$ |

(Haresamudram et al. (2022))

## 5.3 ABLATION STUDY

### 5.3.1 SLOT VARIABILITY

To assess how downstream tasks prioritize different parts of the slot-based representation, we first examine the slot weights learned by the task heads, taken from the first linear layer. As shown in the top row of Figure 4, the weight distributions differ across tasks: some tasks assign noticeably larger weight on a small subset of slots (e.g., Writing and TennisShot), while others distribute weight more evenly. This indicates that relative importance assigned to slots is not uniform across tasks.

To further examine this variability, we perform a slot-masking experiment where we run inference while masking out one slot at a time. The resulting performance drops, shown in the bottom row of Figure 4, also vary across tasks. The slot whose removal causes the largest degradation differs by

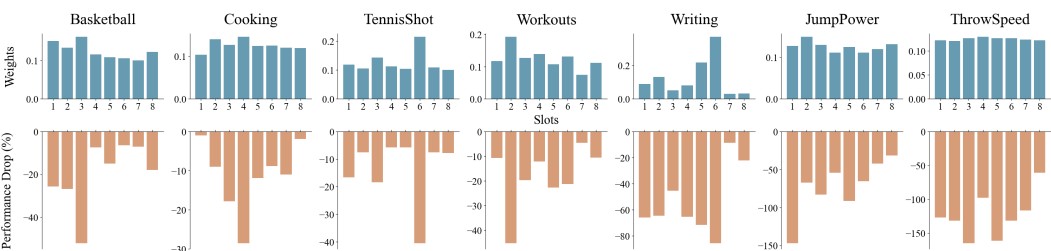

Figure 4: Softmax-normalized slot weights from the first task-head layer (top) and corresponding performance drops when each slot is masked at inference (bottom). Different tasks rely on different subsets of slots, reflected by both their weight patterns and the drops caused by masking each slot.

task, suggesting that reliance on slots differs by task. A complementary t-SNE visualization of the most and least weighted slots is also provided in Appendix A.4.4. Overall, these analyses show that different tasks draw on different slot components within the learned representation.

### 5.3.2 MODEL COMPONENTS

Table 4 shows the mean performance drop when different components of SlotFM are removed, highlighting the importance of each. Additional ablation results on hyperparameters such as the number of bands and the number of slots are included in Appendix A.4.

Removing the bandpass filter and using the original signal directly leads to a performance drop of about 6%. This is because without frequency decomposition, the slots distribute only along the temporal axis but not across frequency ranges. The loss regularizers are also critical, as removing it causes a drop of over 15%, particularly on regression tasks. This indicates that preserving richer information in the slot vectors is essential for strong performance. A visualization of reconstructions with and without our loss regularizers are shown in Appendix A.4.1.

When Slot Attention is removed and a standard autoencoder is trained using our loss regularizers on bandpassed signals, performance drops by about 10%, highlighting the importance of Slot Attention. On the other hand, it still outperforms a default autoencoder trained on the raw signal with MSE loss, showing that bandpass filtering and improved reconstruction losses enhance overall performance. In addition, removing positional embeddings from the encoded frames leaves cross-attention without temporal and spectral context, leading to a performance drop of over 10%.

When slot initializations are sampled from a learned normal distribution, as in the original Slot Attention, each slot captures different information on each pass even for the same input. This prevents the head from focusing on specific slots, reducing downstream performance by over 15%. The self-attention head is a small module added before the MLP head used for downstream tasks. Though it holds less than 10% of the head's total parameter count, it brings about a 9% performance boost on average. Increasing other baselines' head sizes by the same amount does not yield similar improvements, suggesting that this structure can effectively leverage the slot representations.

To further illustrate the effect of each component, Figure 5 shows t-SNE projections of the embeddings for the Workouts dataset under different ablation settings. The full SlotFM model produces

Table 4: Average performance degradation when dropping key components of SlotFM.

|  | Classification tasks | Regression tasks |
|---|---|---|
| w/o Bandpassed input | -7.4% | -6.0% |
| w/o Loss regularizers | -13.2% | -37.1% |
| w/o Slot Attention | -10.0% | -10.0% |
| w/o Soft positional embeddings | -10.5% | -14.0% |
| w/o Fixed slot initialization | -15.9% | -25.0% |
| w/o Self-attention head | -8.0% | -11.1% |

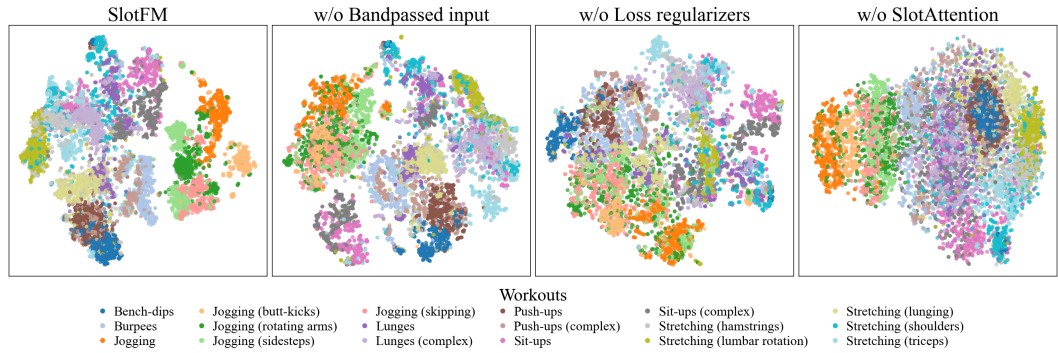

Figure 5: t-SNE visualizations of embeddings for the Workouts task under different ablation settings. The full SlotFM model shows clearer class structure, while removing bandpassed input, loss regularizers, or Slot Attention leads to more mixed embeddings.

clusters with clearer class structure, while removing bandpassed input, loss regularizers, or Slot Attention results in more mixed and overlapping embeddings. These qualitative differences align with the performance drops in Table 4, indicating that each component contributes to preserving discriminative structure in the learned representation.

## 6 DISCUSSION & CONCLUSION

We introduce SlotFM, an accelerometer foundation model trained with Time-Frequency Slot Attention and an improved reconstruction loss, enabling embeddings that capture distinct aspects of the signal while preserving both low- and high-frequency information. Our evaluations on 16 downstream tasks, spanning classification and regression across domains such as gestures, sports, cooking, and transportation, demonstrate that SlotFM generalizes well beyond the scope of most prior foundation models. These results highlight the potential of a single foundation model supporting multi-task inference, offering a practical solution for wearable devices with limited resources. With our open-source code release of both the model implementation and the downstream task setup, we encourage further research toward developing a general foundation model for wearable sensor data.

In this work, we focus on accelerometer data because it is a low-cost sensor widely integrated into wearable devices and provides rich information for diverse applications. Given that other time-series sensors, such as gyroscopes, magnetometers, and photoplethysmography, also contain information across temporal and frequency dimensions, future work could investigate extending our approach to these modalities. Furthermore, while we concatenate slot vectors into an embedding to be used for downstream tasks, future work could explore alternative ways of processing slots. For example, slots could be used as tokens for a larger model, allowing more dynamic interactions between them. Another possibility is to sample slots differently, such as learning an initial slot vector specific to each task that acts as a query for that task alone. These directions could unlock new uses of slot-based embeddings for time-series data.

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

# A APPENDIX

## A.1 TASK DIVERSITY EVALUATION DATASETS

For the Task Diversity Evaluation, all downstream task datasets are resampled to 50 Hz and segmented into non-overlapping 5-second windows to match the data used for pre-training unless indicated otherwise. Units are converted to $m/s^2$, and no additional normalization is applied other than whatever the datasets are published with. The code for dataset parsing will be released.

To account for differences in dataset characteristics and sizes across downstream tasks, we evaluate each task with six configurations: three learning rates ($1 \times 10^{-3}, 5 \times 10^{-4}, 1 \times 10^{-4}$) and two MLP depths (2 or 3 layers). The best-performing configuration is reported for each task and baseline.

**Basketball** is parsed from the Hang-Time HAR dataset (Hoelzemann et al., 2023). We use the five basketball related actions: dribbling, shot, layup, pass, and rebound.

**Cooking** is parsed from the PrISM dataset (Arakawa et al., 2023). Since only the starting timestamp of each action is indicated, we cut a single 5-second window from the start of each action. We use all 17 actions in the dataset.

**Locomotion** is parsed from the RealWorld dataset (Sztyler & Stuckenschmidt, 2016). We use all eight locomotion related activities in the dataset.

**TennisShot** is parsed from the Silent Impact dataset (Park et al., 2024). Since each shot is segmented into 1.5-second windows sampled at 120Hz, we use the data as is without downsampling to 50Hz, and fill the remaining 70 frames by repeating the last value. We use the data of the passive arm and use all six shot classes.

**Transportation** is parsed from the Sussex-Huawei Locomotion and Transportation (SHL) Dataset (Gjoreski et al., 2018). We used the IMU data collected by a phone in the front pocket of the participants. Since there are only 3 participants in the dataset, we evenly distribute consecutive segments of the same label class into 5 folds. We use six classes: still, bike, car, bus, train, and subway.

**Workouts** is parsed from the WEAR dataset (Bock et al., 2024). We use all 18 workout activities in the dataset.

**Writing** is parsed from the HCI Tabletop Gestures dataset (Roggen, 2022). Since each character is written quickly, typically in under 2 seconds, we take 2.5-second segments sampled at 100Hz, filling the remaining frames by repeating the last value. We only use the 'Table' mode data, as the 'Mouse' and 'Slate' modes show insufficient wrist movement to distinguish between classes. We use all 39 character classes in the dataset.

**JumpPower** is parsed from the dataset by White et al. (2022). We use all jumps from all 73 participants, with the *peak power* value used as the regression label.

**NumSteps** is parsed from the dataset by Santos et al. (2022). We use the motion capture 3D coordinates of both ankles to count the number of steps taken within each window.

**ThrowSpeed** is parsed from the dataset by Gençoğlu & Gümüş (2020). As the dataset includes only four participants and does not specify participant IDs, we split all trials into five non-overlapping folds for 5-fold cross-validation.

**WalkDistance** is parsed from the dataset by Santos et al. (2022). We use the motion capture coordinates of the clavicles to track the participant's 2D position and calculate the distance walked within each window.

## A.2 SLOTFM IMPLEMENTATION DETAILS

### A.2.1 PSEUDOCODE

Algorithms 1–3 provide the full pseudocode for our SlotFM architecture. Algorithm 1 describes how an IMU window is decomposed into three frequency bands and encoded into frame features. Algorithm 2 details the Slot Attention refinement module, following Locatello et al. (2020), which

iteratively updates a set of $K$ slot vectors conditioned on the encoded frames. Finally, Algorithm 3 outlines the slot-based band decoder used during pre-training, where per-slot reconstructions are combined using softmax-normalized masks to obtain the reconstructed IMU band signals.

---

**Algorithm 1: IMU band encoding.** A window of IMU data is split into three frequency bands, encoded with band-specific encoders to produce frame features $E$ with $N = 3T$ frames.

---

**Input**    : $X \in \mathbb{R}^{C \times T}$
**Output** : $E \in \mathbb{R}^{N \times D_{\text{slot}}}$
**Params** : BandpassFilters; $\text{Encoder}_{\text{low}}$; $\text{Encoder}_{\text{mid}}$; $\text{Encoder}_{\text{high}}$

$X_{\text{low}}, X_{\text{mid}}, X_{\text{high}} = \text{BandpassFilters}(X)$
$E_{\text{low}} = \text{Encoder}_{\text{low}}(X_{\text{low}})$                              // $\mathbb{R}^{T \times D_{\text{enc}}}$
$E_{\text{mid}} = \text{Encoder}_{\text{mid}}(X_{\text{mid}})$
$E_{\text{high}} = \text{Encoder}_{\text{high}}(X_{\text{high}})$
$E = \text{stack}(E_{\text{low}}, E_{\text{mid}}, E_{\text{high}})$                          // $E \in \mathbb{R}^{N \times D_{\text{enc}}}$

**return** $E$

---

**Algorithm 2: Slot Attention Update.** Frame features $E$ are augmented with positional embeddings, projected to $D_{\text{slot}}$, and mapped to $K$ slots through $U = 3$ iterative attention updates.

---

**Input**    : $E \in \mathbb{R}^{N \times D_{\text{slot}}}$
**Output** : $S \in \mathbb{R}^{K \times D_{\text{slot}}}$
**Params** : $S_{\text{initial}}$; $W_q, W_k, W_v$; GRU; MLP; PositionalEmbedding; LinearProjection; LayerNorm

$E = \text{PositionalEmbedding}(E)$
$E = \text{LinearProjection}(E)$                          // $E \in \mathbb{R}^{N \times D_{\text{slot}}}$
$E = \text{LayerNorm}(E)$
$k = W_k(E),\ v = W_v(E)$
$S = S_{\text{initial}}$
**for** $u = 1 \ldots U$ **do**
  $S_{\text{prev}} = S$
  $S = \text{LayerNorm}(S)$
  $q = W_q(S)$
  $\text{scores} = (qk^{\top})/\sqrt{D_{\text{slot}}}$                          // $\mathbb{R}^{K \times N}$
  $\text{attn} = \text{softmax}(\text{scores, axis=slots})$
  $\text{updates} = \text{attn} \cdot v$                          // $\mathbb{R}^{K \times D_{\text{slot}}}$
  $S = \text{GRU}(\text{state} = S_{\text{prev}}, \text{inputs} = \text{updates})$
  $S = S + \text{MLP}(\text{LayerNorm}(S))$

**return** $S$

---

**Algorithm 3: Slot decoding.** $K$ slot vectors are decoded into band-specific signals and masks. The masks are normalized across slots and used as weights to reconstruct each band.

---

**Input**    : $S \in \mathbb{R}^{K \times D_{\text{slot}}}$
**Output** : $\hat{X}_{\text{low}}, \hat{X}_{\text{mid}}, \hat{X}_{\text{high}} \in \mathbb{R}^{C \times T}$
**Params** : $\text{Decoder}_{\text{low}}$; $\text{Decoder}_{\text{mid}}$; $\text{Decoder}_{\text{high}}$

**for** $k = 1 \ldots K$ **do**
  $R_{\text{low}}[k], M_{\text{low}}[k] = \text{Decoder}_{\text{low}}(S[k])$                          // $R : \mathbb{R}^{C \times T}, M : \mathbb{R}^{1 \times T}$
  $R_{\text{mid}}[k], M_{\text{mid}}[k] = \text{Decoder}_{\text{mid}}(S[k])$
  $R_{\text{high}}[k], M_{\text{high}}[k] = \text{Decoder}_{\text{high}}(S[k])$
$M_{\text{low}} = \text{softmax}(M_{\text{low}}, \text{axis=slots})$
$M_{\text{mid}} = \text{softmax}(M_{\text{mid}}, \text{axis=slots})$
$M_{\text{high}} = \text{softmax}(M_{\text{high}}, \text{axis=slots})$
$\hat{X}_{\text{low}} = \sum_{k=1}^{K} M_{\text{low}}[k] \odot R_{\text{low}}[k]$
$\hat{X}_{\text{mid}} = \sum_{k=1}^{K} M_{\text{mid}}[k] \odot R_{\text{mid}}[k]$
$\hat{X}_{\text{high}} = \sum_{k=1}^{K} M_{\text{high}}[k] \odot R_{\text{high}}[k]$

**return** $\hat{X}_{low}, \hat{X}_{mid}, \hat{X}_{high}$

---

Table 5: Loss weights and configurations for each frequency band.

| Band | Losses Used | $\alpha$ (MSE) | $\beta$ (SSIM) | $\gamma$ (MS-STFT) | Window Size |
|------|-------------|----------------|----------------|--------------------|-------------|
| Low | MSE only | 1 | 0 | 0 | – |
| Mid | MSE, SSIM, MS-STFT | 1 | 100 | 0.1 | 25 |
| High | SSIM, MS-STFT | 0 | 50 | 0.1 | 10 |

### A.2.2 TRAINING HYPER-PARAMETERS

We train using the Adam optimizer with a learning rate of 0.0001. The hidden dimension is set to 256, and the number of slot update iterations is fixed at three. At each batch, data from four participants are loaded, where for each participant, 128 windows are sampled with probabilities proportional to the standard deviation of their signals. This results in a batch size of 512.

### A.2.3 POSITIONAL EMBEDDING

As in the original Slot Attention work (Locatello et al., 2020), we augment the encoded features with 2D positional embeddings right before passing them into Slot Attention, since Slot Attention is invariant to the order of the input elements. A $W \times H \times 4$ tensor is created, where $W$ is the number of frequency bands and $H$ is the number of frames per band, with each of the four channels representing a linear gradient in one of the cardinal directions $(x, y, 1 - x, 1 - y)$.

### A.2.4 LOSS REGULARIZERS

We combine three loss functions, MSE, SSIM and MS-STFT, each weighted by coefficients $\alpha$, $\beta$, and $\gamma$, respectively. Because the three frequency bands differ substantially in their signal characteristics, we assign distinct weights to each band. Note that SSIM is given a much larger weight since its loss value is bounded between 0 and 1, whereas MSE and MS-STFT are unbounded and naturally operate on a larger scale.

The loss weights were chosen by first matching the scales of the losses during the initial epochs and then selecting among a few coarse candidate values that yielded stable training and consistent reconstruction quality. For the low-frequency band, which captures slow, large-magnitude movements, we apply only MSE to preserve absolute shifts from the zero axis. For the mid-frequency band, which contains more nuanced variations, we use all three losses with weights $\alpha = 1$, $\beta = 100$, and $\gamma = 0.1$, and a window size of 25 frames. For the high-frequency band, which captures short-period fluctuations, we use only SSIM and STFT with weights $\beta = 50$, $\gamma = 0.1$, and a smaller window size of 10 frames. Table 5 summarizes the final configuration.

## A.3 SSL BASELINE IMPLEMENTATION DETAILS

We outline the implementation details for the five baseline SSL methods compared in the Task Diversity Evaluation: Autoencoder, Masked Autoencoder, SimCLR, Augmentation Prediction, and RelCon. Each method was adapted from prior implementations but modified to ensure a fair comparison. The data, dataloading mechanism, batch size, and training epochs were kept identical to SlotFM. The encoder architecture was also standardized, using a ResNet with 12 blocks that preserves the temporal dimension, followed by Global Average Pooling and a linear layer to produce a 256-dimensional embedding. This results in an encoder with 4.8M parameters, matching SlotFM. Additional network components specific to each SSL approach were added consistent with the original work. Where necessary, design choices were explored to identify the best-performing configurations.

**Autoencoder** follows the convolutional autoencoder from Haresamudram et al. (2019). The ResNet backbone generates embeddings that are decoded into the reconstructed signal using a mirrored ResNet decoder. The model is trained with the Adam optimizer and MSE loss between the original and reconstructed signal.

**Masked Autoencoder** extends the autoencoder by randomly masking parts of the raw input before encoding. After experimenting with different masking ratios (20%, 50%, 80%) and segment sizes

Table 6: Parameter sizes of SlotFM and baseline models. Pretext-specific components are included only during pre-training.

| Method | Encoder Params | Pretext / Decoder Params | Total Params |
|---|---|---|---|
| SlotFM | 4.8M | 4.8M (decoder) | 9.6M |
| Autoencoder | 4.8M | 4.8M (decoder) | 9.6M |
| Masked Autoencoder | 4.8M | 4.8M (decoder) | 9.6M |
| Augmentation Prediction | 4.8M | 2K (pretext classifier) | 4.8M |
| SimCLR | 4.8M | 80K (projection head) | 4.9M |
| RelCon | 4.8M | 80K (projection head) + 59K (distance network) | 4.9M |

(10, 25, 50 frames per segment), we finalized on masking 50% of the input by dividing each window into 10 segments and randomly selecting 5 to mask per epoch. The model is trained with the Adam optimizer and MSE loss over the full signal.

**Augmentation Prediction** follows the official implementation of Yuan et al. (2024), where the encoder is trained to classify which augmentation was applied to the input. The same ResNet backbone is used, with a separate linear classifier for each augmentation, making it a binary prediction task per augmentation. Following Yuan et al. (2024), random shuffle and rotation are applied for invariance, while the tasks used for prediction are timeflip, permute, and timewarp. Each augmentation is applied with 50% probability. The model is trained with the Adam optimizer and cross-entropy loss.

**SimCLR** follows the official contrastive learning implementation of Tang et al. (2020). Two augmented views of each input window are encoded by the ResNet backbone and projected through a two-layer MLP head into a 64-dimensional embedding. The model is trained with the NT-Xent loss with a temperature of 0.1, with positive and negative pairs drawn from within the batch. After experimenting with different augmentation combinations from the available eight, we finalized on random scaling and time warping, each applied with a probability of 0.3. The model is optimized with SGD (momentum 0.9, weight decay $1 \times 10^{-5}$) and a cosine annealing learning rate scheduler.

**RelCon** follows the official implementation of Xu et al. (2025). The Learnable Distance Measure uses the same architecture as in the original work and is trained with MSE loss between the input and reconstructed signal. In each batch, two of the eight available augmentations are chosen at random and applied to the input. The encoder of the main backbone is replaced with our ResNet-12, and the embeddings projected into a 64-dimensional space. The model is trained using the original Relative Contrastive Loss and the Adam optimizer.

Table 6 summarizes the number of trainable parameters for SlotFM and all baselines. Note that all methods use the same encoder size, and only the decoder or other pretext-specific components used during pre-training differ in parameter count.

## A.4 FURTHER ABLATION STUDIES

### A.4.1 LOSS REGULARIZER

Figure 6 compares the reconstructions of SlotFM with and without the proposed loss regularizers, $\mathcal{L}_{\text{SSIM}}$ and $\mathcal{L}_{\text{MS-STFT}}$. When trained with only MSE, the model captures the overall trend of the

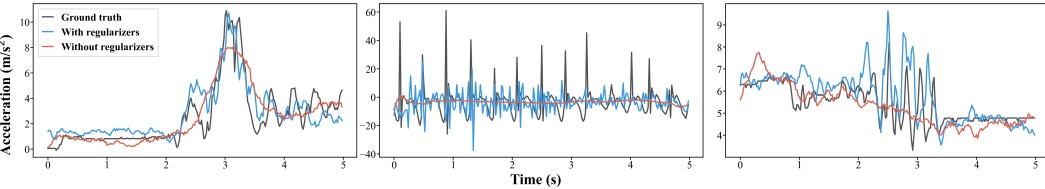

Figure 6: Signal reconstruction with and without our loss regularizers. Using the regularizers better preserves higher-frequency details. Only one channel is displayed.

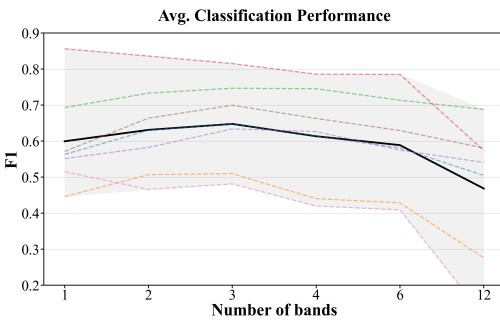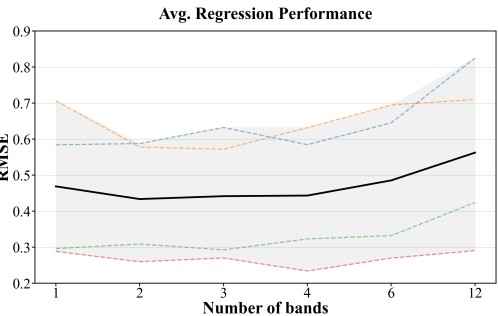

Figure 7: Average performance on classification and regression tasks when the input signal is band-passed into different numbers of bands. The solid black line shows the overall average, and the dotted colored lines show the performance of each individual task.

Table 7: Cutoff frequency boundaries for each band configuration.

| # Bands | Cutoff Boundaries (Hz) |
|---------|------------------------|
| 1 | 0-25 |
| 2 | 0-1, 1-25 |
| 3 | 0-1, 1-4, 4-20 |
| 4 | 0-1, 1-4, 4-10, 10-25 |
| 6 | 0-1, 1-2, 2-4, 4-6, 6-10, 10-25 |
| 12 | 0-1, 1-2, 2-3, 3-4, 4-5, 5-6, 6-8, 8-10, 10-12, 12-14, 14-16, 16-25 |

signal but fails to reproduce higher-frequency fluctuations. In cases with sharp peaks, as shown in the second example, the reconstruction completely misses these features. In contrast, when the loss regularizers are applied, although the reconstructions still do not detect sharp peaks to the exact same magnitude, the general patterns of these sharp transitions remain are preserved.

### A.4.2 NUMBER OF BANDS

Figure 7 shows the effect of the number of bands used as input for the model. The solid black line shows the average across tasks, while the dotted colored lines show individual task performances. In calculating the average, the RMSE values for the JumpPower and ThrowSpeed tasks are divided by 10 to match the scale of the other regression tasks.

For each configuration, the input signal is processed through a bandpass filter to decompose it into a given number of bands. Table 7 shows the cutoff boundaries for each configuration. Since most of the motion in the signals lies in the lower frequencies, typically under 10 Hz, we allocated finer granularity to the lower frequency ranges. Following our model design, where each band has a separate encoder to account for differences in patterns and shapes across frequency ranges, we adjusted the number of encoder layers so that the total model size remained constant. This means that the single-band configuration uses the largest encoder with 12 ResBlocks, while the 12-band configuration uses only one ResBlock per band.

The results show that while decomposing the signal into multiple bands improves performance, more bands do not necessarily lead to better performance. For both classification and regression tasks, performance begins to degrade from 6 bands onward. This is likely because splitting into too many bands makes each band narrower and less informative, and the smaller encoders assigned to each band may not have enough capacity to capture meaningful features. Using 12 bands leads to a significant drop in performance, since higher frequencies contain less information, and dedicating individual bands to these ranges does not contribute anything meaningful.

### A.4.3 NUMBER OF SLOTS

Figure 8 shows the effect of the number of slots on classification and regression performance. The size of each slot embedding is adjusted so that the total embedding size remains constant. For

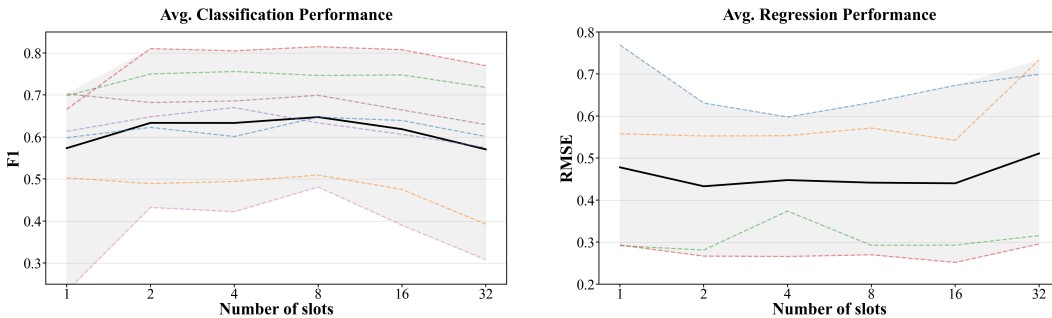

Figure 8: Average performance on classification and regression tasks with different numbers of slots. The solid black line shows the overall average, and the dotted colored lines show the performance of each individual task.

example, with 2 slots the slot dimension is 128, while with 32 slots the slot dimension is 8. Overall, the performance is relatively stable across different slot numbers. For classification, 8 slots give the best performance, while larger slot numbers lead to a drop, especially at 32. For regression, performance is mostly stable across slot numbers, but 32 shows a clear degradation.

### A.4.4 T-SNE ANALYSIS OF SLOT USAGE

Figure 9 shows t-SNE projections of the two most and two least weighted slots for three tasks. The slots that exhibit the strongest separation differ by task: a slot that provides clear structure for one task may be among the least informative for another. The most weighted slots generally produce more distinct clusters, while the least weighted slots appear more mixed.

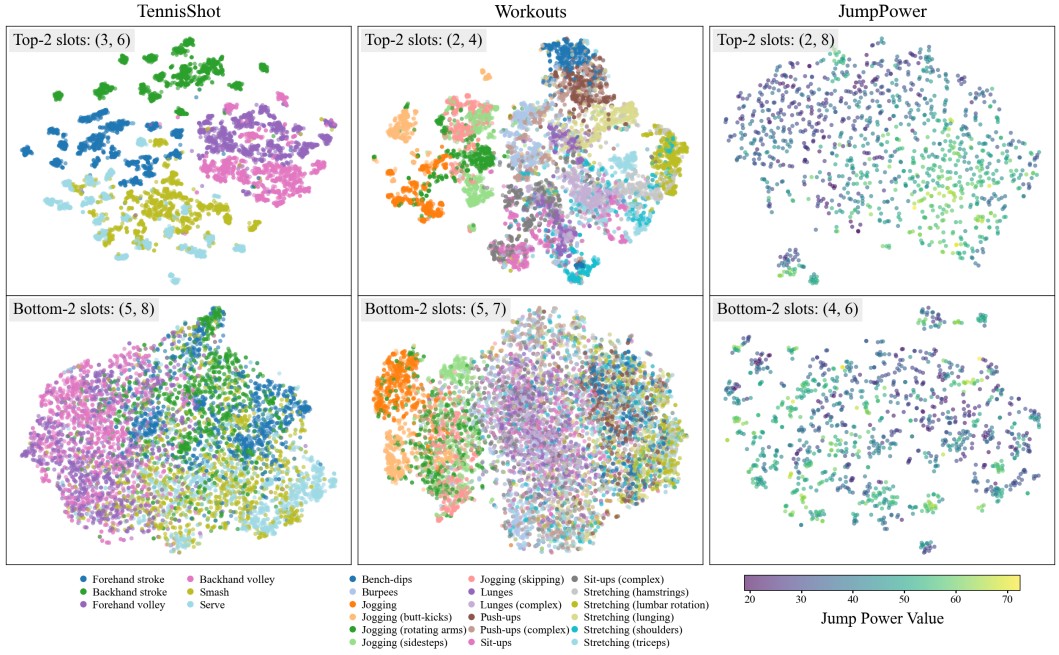

Figure 9: t-SNE visualizations of embeddings from the two most weighted slots (top) and two least weighted slots (bottom) for three tasks. The slots that show stronger class- or value-related structure vary by task, indicating that different tasks rely on different subsets of slots.

### A.5 LLM Usage Disclosure

In preparing this manuscript, we used LLMs solely as a writing aid to improve grammar, phrasing, and formatting (e.g., LaTeX macros, table layout). LLMs were not used for experiments, analyses, or substantive research content. The authors take full responsibility for the final manuscript.

