# OpenReview forum: "SlotFM: A Motion Foundation Model with Slot Attention for Diverse Downstream Tasks"
_ICLR.cc/2026/Conference — Submitted to ICLR 2026_

### Official Review · Reviewer_biqt · 2025-10-30

**Soundness:** 3
**Presentation:** 3
**Contribution:** 2
**Rating:** 4
**Confidence:** 4

**Summary:**

This paper proposes a time-frequency slot attention-based motion foundation model for accelerometers, aiming to generalize across multiple downstream tasks (HAR, pose, gesture, motion, etc.). The paper combines frequency domain decomposition with a slot-based self-attention structure and introduces two regularization losses, SSIM and MS-STFT, to improve signal reconstruction quality.

**Strengths:**

This paper is well organized.
The overall logic is clear.

**Weaknesses:**

Overall, this is a reasonably good paper; however, while the motivation is to establish a motion foundation model that generalizes across multiple accelerometer tasks, the authors do not clearly explain why existing multi-task SSL models fail to generalize to the same task set. Furthermore, the theoretical basis for the slot attention mechanism, originally used for object decomposition in images, (such as the equivalence of time-frequency structures) for its transfer to time-series signals is not sufficiently demonstrated. A clear hypothesis or quantitative motivation is lacking to explain why the "slot-based" method has a natural advantage in IMU signals.

Weaknesses:
1.	The paper only pre-trained on CAPTURE-24, while the test tasks differed greatly from the original domain. Although the results showed good generalization, there were no explicit cross-domain validation (cross-person or cross-device) experiments.
2.	Reporting only the average decrease percentage in ablation experiments is insufficient to validate the independent contribution of the modules. It is recommended to include interpretability visualizations or t-SNE results to demonstrate the separability of slots across different tasks.
3.	Although the authors claim to have open-sourced the code and task settings, they haven't provided details on hyperparameters in the paper, nor have they included training time, memory usage, or training stability reports for comparative experiments in the appendix. In the experimental setup, SlotFM uses a fixed number of slots, a self-attention head, and a three-way ResNet encoder, making the number of parameters seem likely to be significantly larger than the baseline (vs. single ResNet). While the authors state that the total number of parameters is "similar," they haven't provided a precise parameter comparison table. Considering the model requires 500 epochs of training on a 4×V100, the actual computational cost is high, potentially impacting reproducibility and practical applications.

**Questions:**

1.	Slot Attention's cross-attention is insensitive to the input order, so in this work the authors introduce 2D positional embedding, but do not prove that this positional encoding can preserve the temporal dependence of continuous signals.
2.	Since frequency band decomposition has already separated low/high frequency information, is the gain of Slot Attention in reallocating slots significant, or is there any redundancy?
3.	The loss function is defined as a combination of three loss functions (MSE, SSIM, MS-STFT). However, SSIM is essentially a similarity measure (range 0-1), while MSE and STFT loss appear to be unbounded energy errors, with a huge difference in magnitude. The authors set β=100 and γ=0.1 based solely on experience without providing scale normalization or sensitivity analysis. Applying such a large difference in loss combination may also lead to insufficient generalization ability of the method, resulting in optimization instability or bias towards specific frequency bands, making it extremely difficult to apply this method to other fields or different datasets.

---

> ### Author Response · Authors · 2025-11-20
> **Response to Reviewer biqt (part 1)**
>
> Thank you for the constructive review of our work. Prior multi-task SSL methods for IMU data, such as SimCLR and Augmentation Prediction, are sensitive to the chosen augmentations. This can bias the learned representation toward invariances that are favorable for HAR while suppressing information needed for a broader range of motion-related tasks. We therefore adopt reconstruction-based approaches, which preserve the full input information. However, standard autoencoders compress each window into a single embedding, limiting their ability to capture the diverse temporal and spectral patterns present in IMU signals. Slot Attention offers a way to decompose the input into multiple vectors that can specialize in different tempo–spectral regions. The strong generalization across our downstream results supports the usefulness of this decomposition. We have also clarified this motivation more explicitly in the second paragraph of the Introduction section.
>
> ### **Weakness 1 (Cross-domain validation)**
>
> Thank you for raising this point. Regarding cross-domain behavior, although we did not explicitly frame them as such, all downstream tasks already differ from the pre-training domain in several ways, including device type, participant pool, sampling rate (e.g., TennisShot & Writing tasks), and sensor placement (e.g., waist/leg in the HAR benchmark). Downstream training also uses participant-disjoint folds, which provides cross-person validation as noted in Section 4.2.1. However, we are not fully certain we understood the concern, and would greatly appreciate clarification if our response has not fully addressed it.
>
> ### **Weakness 2 (Ablation visualization and interpretability)**
>
> Thank you for the suggestion. We have expanded the interpretability analysis by adding t-SNE visualizations of slot embeddings under different ablation settings (now included after Table 4). These plots show that removing core components such as bandpassed input, loss regularizers, or Slot Attention results in more mixed and overlapping clusters, which is consistent with the performance drops observed in the ablation table. In addition, Figure 5 illustrates reconstruction quality with and without the loss regularizers, where better reconstruction preserves more signal structure and supports stronger representations. We hope these additions provide clearer insight into the independent contributions of each module.
>
> ### **Weakness 3 (Training details, efficiency, and parameter comparison)**
>
> We appreciate the reviewer’s concern regarding computational cost and reproducibility. SlotFM’s encoder has 4.8M parameters, which is equal to the encoder size used in our baselines. This is because we  configured each per-band encoder to one-third of the size of the baseline encoder (4 residual blocks per band vs. 12 residual blocks total), as described in Sections 4.1 and 4.2.1. As with Autoencoder and MAE baselines, SlotFM includes an additional decoder during pre-training (~4.8M parameters), while other methods include pretext-specific components (e.g., distance network for RelCon, projection head for SimCLR). We have included a parameter comparison table in the Appendix to strengthen reproducibility.
>
> Regarding training time, under our configuration, all baselines except RelCon required slightly under one hour to train for 500 epochs. RelCon, due to the cost of the Relative Contrastive Loss computations, required roughly six hours. SlotFM trained in roughly three hours, which is longer than most baselines but still well within a scalable and practical range for SSL pre-training.
>
> Finally, we reiterate that the full model and training code will be publicly released upon acceptance.

---

> ### Author Response · Authors · 2025-11-20
> **Response to Reviewer biqt (part 2)**
>
> ### **Question 1 (Positional encoding for temporal structure)**
> Positional embedding was used in the original Slot Attention algorithm because cross-attention is order-invariant. We follow the same design and apply the identical 2D positional encoding on axes corresponding to frequency band and time rather than spatial coordinates. This allows the model to preserve the temporal dependence of the signal while also distinguishing the different frequency bands. To support this, we added an ablation showing that pre-training without positional embedding leads to an average performance drop of more than 10% for both classification and regression downstream tasks.
>
> ### **Question 2 (Band decomposition and slot allocation)**
> Band decomposition provides input signals separated into distinct frequency ranges. Slot Attention then learns how to allocate attention across both the frequency and temporal dimensions within those bands. These mechanisms therefore play complementary roles rather than being redundant. The ablation that removes Slot Attention results in a clear performance drop, confirming that this learned specialization contributes independently to the model’s effectiveness.
>
> ### **Question 3 (Loss weighting and stability)**
> We agree that balancing multi-term losses is important. Since these losses operate on different scales, we weighted them to bring their magnitudes closer, as standard practice when combining multiple objectives. Rather than performing a full systematic sweep, we first matched the scales based on the losses from the initial epochs and then ran a small set of exploratory trials around those values. We selected the final weights based on which setting produced the most stable training and the best overall reconstruction quality across the three bands. We have clarified this in the Appendix A.2.4.

---

### Official Review · Reviewer_D2U4 · 2025-10-31

**Soundness:** 3
**Presentation:** 3
**Contribution:** 3
**Rating:** 6
**Confidence:** 5

**Summary:**

This paper presents a novel foundation model for a wide variety of motion related tasks with sota performance. Additionally, they proposed a novel time-frequency slot attention methodology adapted for time-series ssl that exploits specific time and frequency information.

**Strengths:**

* Well formulated time-frequency slot attention mechanism that exploit time-series specific properties for the setting.
* Thoughtful loss regularizers that address key problems
* Slot reconstruction visualization offers interpretable insights into specific learned slots
* Diverse downstream tasks and datasets that cover a wide range of motion-related tasks, greatly expanding upon prior work

**Weaknesses:**

* The model architecture in 3.2 is not super clear, and generally requires that readers have an exact understanding of slot attention. as another example, a GRU cell is mentioned, but not seen in Figure 3.  the paper could perhaps benefit from including a mathematical formulation of architecture. this is not a strict necessity, but something to improve clarity in 3.2 would be helpful.
* Figure 4 with weights per slot distribution is not well explained. Why are slots 4/5 the beginning of the signal? This analysis on slot weights seems underexplored and underexplained.

**Questions:**

* This is not a strong criticism, but something I would appreciate further discussion on is needing to calculate multi-scale STFT for every instance and the time complexity associated with this. this seems to me to be very slow, thus making this method not necessarily scalable for large-scale pre-training. once again, this does not kill the idea of the paper, but i would appreciate better understanding of the limitations in general.
* Haresumadram et al has not released their exact train/val/test splits for all datasets and giving the numbers are directly quoted from that paper, this should be addressed more clearly. The RelCon codebase does have the splits for some of the datasets, but notably not FoG nor USC-HAD. How were the exact splits determined? If they were not exactly the same, I recommend reaching out to Haresumadram to obtain the exact splits during rebuttal or final version.
* A code release is mentioned, but it is not a part of the current submission. Can this be clarified?

---

> ### Author Response · Authors · 2025-11-20
> **Response to Reviewer D2U4**
>
> Thank you for the helpful and constructive feedback. We appreciate the positive assessment of our contributions and have revised the paper to strengthen clarity and address your concerns.
>
> ### **Weakness 1 (Clarifying architecture details in Section 3.2)**
>
> Thank you for the suggestion. We have now included detailed pseudocode for the Time-Frequency Slot Attention module in the Appendix. This explicitly covers all components involved in the update mechanism, including the GRU cell and MLP. Since these elements follow the original Slot Attention formulation, we had omitted them from the figure to avoid clutter. To avoid confusion, we renamed “Cross Attention” in Figure 3 to “Slot Attention Update” to better reflect that the full iterative Slot Attention process is applied. We believe these updates improve clarity without overloading the main architecture figure.
>
> ### **Weakness 2 (Clarifying interpretation of slot weights)**
>
> We appreciate this comment and have revised the explanation to improve clarity. During pre-training, the slots naturally specialize to different tempo-spectral regions of the input. In our model, slots 4 and 5 learned to focus on the beginning of the signal window, as their decoded reconstructions reproduce that early portion of the input while remaining inactive elsewhere.
>
> To further strengthen our analysis on slot emphasis, we included two additional results (also referenced in our response to Reviewer iJWj
> - Slot masking experiments – Masking different slots leads to task-specific performance drops, indicating that distinct tasks rely on different slots
> - Slot-wise embedding visualizations – t-SNE plots of most vs. least weighted slots show large variation in separability across tasks
>
> Together, these provide stronger empirical grounding for the conclusion that some slots capture more task-relevant information than others depending on the task.
>
> ### **Question 1 (Computational cost of MS-STFT)**
>
> Thank you for raising this point. Computing MS-STFT has complexity O(N log L), where N is the window length and L is the FFT size. In our case, both are very small (5-second IMU windows at 50 Hz and FFT sizes of 16–128), so the overhead remains negligible. Similar multi-scale STFT losses are used in audio models on much longer signals, suggesting that the cost is manageable in practice. In our setting, it is even smaller due to the short window length. In fact, the added training time from using MS-STFT was less than half a second per epoch (<1% of total training time). Furthermore, the STFT of the ground truth can also be computed once and reused during training, which would further reduce computational cost.
>
> ### **Question 2 (HAR split reproducibility)**
>
> Haresamudram et al. report using 5-fold cross-validation but do not release the exact splits. Similarly, the RelCon codebase implements randomized 5-fold generation, but doesn’t provide the original folds. We therefore used the RelCon implementation to generate splits for the three datasets it supports (Motionsense, HHAR, PAMAP2) and applied the same protocol to FoG and USC-HAD for consistency. As a result, although the exact participant assignments may differ, we follow the same 5-fold protocol, which should be relatively stable under small variations in fold assignment. We have reached out to the authors to request the original splits and will update our results accordingly if they are made available. For now, we have also clarified in the paper that the splitting protocol is the same but the exact folds may differ.
>
> ### **Question 3 (Code availability)**
> We confirm that the full pre-training and downstream evaluation code will be publicly released upon acceptance.

---

> > ### Comment · Reviewer_D2U4 · 2025-11-24
> >
> > Thanks for the reply. All of my concerns have been clearly addressed, contingent on the release of full pre-training and downstream evaluation code. I have raised my score accordingly, great work!

---

### Official Review · Reviewer_iJWj · 2025-11-01

**Soundness:** 3
**Presentation:** 3
**Contribution:** 3
**Rating:** 8
**Confidence:** 3

**Summary:**

The authors propose a foundation model for accelerometer tasks called SlotFM that is able to generalize across diverse downstream tasks through self-supervised learning.

It exploits Slot Attention to encode each signal into multiple vectors or slots across several time intervals and three frequency bands, which are used to reconstruct the input signal during training. These slots are used during inference as embeddings for accomplishing downstream tasks. The authors introduce two regularizers to capture local structures and frequency patterns which encourage preservation of the fine-grained details during reconstruction.

The foundation model achieved state-of-the-art results on 13 downstream tasks, and almost-top results for another 3 tasks, showing the ubiquity of the learned slots for diverse range of problems. Surprisingly, in almost all cases, it beat or is just as good as models trained by supervised learning on smaller datasets

**Strengths:**

**Significance**
* Experiments show that self-supervised foundation models can learn embeddings which are useful for a large variety of downstream accelerometer tasks.

**Originality**
* As far as I can tell, this is the first time Slot Attention (originally for computer vision) is applied to accelerometer data
* Improved reconstruction results from two novel regularizers

**Clarity**
* The model architecture and experimental methods are clearly described

**Quality**
* State-of-the-art results across almost all self-supervised methods
* Almost state-of-the-art results when compared with supervised methods

**Weaknesses:**

* The iterative slot attention mechanism is described in both Section 3.1 and 3.2. Even then, it is hard to imagine how this mechanism is actually implemented without reading the original slot attention paper. I am also confused about the implementation of signal reconstruction during pretraining that is describe at the end of Section 3.2. Perhaps the pseudocodes can be provided in the Appendix.

* The weights per slot in Fig 4 seem quite uniform across slots to me, for all tasks except Writing. I do not think they warrant the discussion at the end of Section 5.1,  "The weight distribution among slots differs between tasks, with some tasks emphasizing particular slots while others assigning similar weights to all slots," nor does it warrant the observation at the end of the Introduction, "Finally, analysis of head weights shows that certain slots are emphasized more than others depending on the task, demonstrating the adaptability of slot-based embeddings." More empirical proof is needed, or the statements should be toned down.

**Questions:**

1. Strangely, the task accuracies do not scale with the number of frequency bands or the number of temporal intervals, probably because of overfitting. This seems to suggest that some kind of regularization can help. Do you have any insight/analysis of what is going on here?

2. From Fig 2, it seems that there are 3 x 8 = 24 "slots" (spatial-temporally) but in Fig 4, there are only 8 weights for 8 slots (temporally). Do you consider the combination of low-middle-high band components in one time-interval as one slot? Why not allow the weighted sum of the band components for downstream tasks, so we have 24 weights rather than 8?

---

> ### Author Response · Authors · 2025-11-20
> **Response to Reviewer iJWj**
>
> Thank you for your thoughtful and constructive feedback. We appreciate your positive assessment of the work and have revised the manuscript to address your comments.
>
> ### **Weakness 1 (Clarifying Slot Attention and Reconstruction)**
>
> We agree that Slot Attention and the reconstruction mechanism may be difficult to follow without the original paper. We have added three pseudocode listings in the Appendix that cover the entire pre-training pipeline and clarify implementation details. We have also refined Section 3.2 for clarity and to ensure notations are fully consistent with the pseudocode. Furthermore, the full training and model code, including our reconstruction implementation, will be publicly released upon acceptance. We believe this will further support reproducibility and help readers fully understand our methodology.
>
> ### **Weakness 2 (Strengthening Slot Importance Analysis)**
>
> Thank you for pointing out the need for stronger empirical support. We have expanded our analysis with two additional experiments (Section 5.3.1. and Appendix A.4.4):
> - Slot masking study – For each downstream task, we repeatedly ran inference while masking one slot at a time. The slot whose removal caused the largest performance degradation differed across tasks, indicating that different tasks rely on different slot-specific information.
> - Slot-wise embedding visualization – We added t-SNE projections of the most and least weighted slots across several downstream tasks. These visualizations show that slot-level separability differs by task, suggesting that some slots capture more discriminative information than others depending on the task.
>
> To more accurately reflect these findings, we revised the original claim to state that reliance on slots differs by task.
>
> ### **Question 1 (Scaling with number of bands / slots)**
>
> Thank you for the careful observation. As noted in Appendix A.4.2 and A.4.3, the total number of model parameters is held constant in these ablations. When increasing the number of frequency bands, we reduce the encoder size per band, and when increasing the number of slots, we reduce the embedding dimension per slot. This ensures a fair comparison across different hyperparameter choices, since otherwise larger numbers of bands or slots would simply increase model capacity and confound the effect of the change itself. Under this fixed-capacity setup, performance peaks at moderate settings rather than increasing monotonically. If total capacity were allowed to grow, larger configurations would likely perform better because they contain more parameters.
>
> ### **Question 2 (Why 8 slots instead of 24 band-slots)**
> Figure 2 illustrates 24 reconstructed signals, but these correspond to 8 slot vectors. During pre-training, each slot vector (dimension 32) is decoded into three signals, one for each band, which is why there are three signals under each “Slot X”. During downstream inference, the decoder is not used and only the 8 slot vectors are passed to the task head, which is why Figure 4 contains 8 attention weights. We hope this is clearer now with the refinements made to Section 3.2 and the added pseudocode in response to Weakness 1.

---

### Author Response · Authors · 2025-12-02
**Response to all reviewers and AC**

We thank all reviewers and the ACs for their efforts in reviewing our paper. We have addressed all concerns and questions raised by the reviewers and have strengthened our submission by incorporating their feedback.

Reviewer D2U4 responded to our rebuttal and confirmed that their concerns were clearly addressed. Before the scores were reverted, they had raised their score from 6 to 8. We also confirm that the full pre-training and downstream evaluation code will be released upon acceptance.

Reviewers iJWj and biqt did not respond before the discussion was halted, but we believe our rebuttal fully addresses the points they raised.

We hope our responses provide clarifications to the public and support the AC’s assessment of the submission.

---

> ### Author Response · Authors · 2025-12-02
> **Summary of revisions**
>
> Here is a summary of the major revisions made to the manuscript based on the reviewers’ feedback:
> 1. **Section 3.2 and Appendix A.2.1** – Revised the model architecture description and added pseudocode for the full pre-training process.
> 2. **Section 5.3.1** – Added a new subsection with expanded analysis of slot weights and importance, and included t-SNE visualizations of slots in Appendix A.4.4.
> 3. **Figure 5** – Added a t-SNE visualization of the embeddings to support the model component ablation study.
> 4. **Table 6** – Added parameter counts for SlotFM and all baseline models.
> 5. **Table 4** – Added ablation results for “w/o Soft positional embeddings” to show the effect of positional embeddings.

---

### Meta-Review · Area_Chair_owuS · 2026-01-07

**Summary:**

This paper (1) extends Slot Attention to "Time-Frequency Slot Attention" which separates data into three bands according to time frequency and then applies slot attention, (2) incorporates two regularisers that have existed in image processing and audio processing, and (3) applies the method to accelerometer data. Experiments show advances of the method.

**Reviewer Concerns:**

The reviewers raised concerns on (1) slot attention mechanism, (2) experimental implementation, including code, (3) computing cost, (4) motivation of the method, etc. Some of them have been duly addressed, while concerns on the motivation still remain.

After reading the paper, I have concerns about (1) the algorithmic/methodological contribution is limited. The novelty of this paper lies mostly in separating the raw signal into three parts according to time frequency. The claimed contributions in regularisers are largely borrowing existing technologies from other areas, without significant synergy here. I thus don't think this paper suits ICLR well, and encourage the authors to try a more applied venue. (2) It would be problematic to call the proposed model a "foundation model", given that it has only 4.8 million parameters and does not report any generalisable or emergent capabilities.

**Reviewer Scores:**

The reviewers' score could be slightly increased.

---

### Decision · Program_Chairs · 2026-01-26

Reject